# The Incidence of Myasthenia Gravis in the Province of Ferrara, Italy, in the Period of 2008–2022: An Update on a 40-Year Observation and the Influence of the COVID-19 Pandemic

**DOI:** 10.3390/jcm13010236

**Published:** 2023-12-30

**Authors:** Annibale Antonioni, Emanuela Maria Raho, Domenico Carlucci, Elisabetta Sette, Riccardo De Gennaro, Jay Guido Capone, Vittorio Govoni, Ilaria Casetta, Maura Pugliatti, Enrico Granieri

**Affiliations:** 1Unit of Neurology, Department of Neurosciences and Rehabilitation, University of Ferrara, 44121 Ferrara, Italy; annibale.antonioni@edu.unife.it (A.A.); emanuelamaria.raho@edu.unife.it (E.M.R.); i.casetta@unife.it (I.C.); pglmra@unife.it (M.P.); 2Doctoral Program in Translational Neurosciences and Neurotechnologies, Department of Neurosciences and Rehabilitation, University of Ferrara, 44121 Ferrara, Italy; 3Unit of Neurology, Interdistrict Health Care Department of Neurosciences, S. Anna Ferrara University Hospital, 44124 Ferrara, Italy

**Keywords:** myasthenia gravis (MG), incidence, neuroepidemiology, autoimmune diseases, COVID-19

## Abstract

Myasthenia gravis (MG) is the most common neuromuscular junction disorder. We evaluated the MG incidence rate in the province of Ferrara, Northern Italy, over two time frames (2008–2018 and 2019–2022, i.e., the COVID-19 pandemic) and considered early-onset (EOMG), late-onset (LOMG), and thymoma- and non-thymoma-associated MG. Moreover, in the second period, we assessed its possible relationship with SARS-CoV-2 infection or COVID-19 vaccination. We used a complete enumeration approach to estimate the MG incidence and its temporal trend. For the period of 2008–18, 106 new cases were identified (mean incidence rate 2.7/100,000 people). The highest rates were observed for the over-70 age group and in rural areas, with 17% of thymoma-associated MG. During the COVID-19 period, 29 new cases were identified (average incidence rate 2.1/100,000 people), showing a marked (though not statistically significant) decrease in the mean annual incidence compared to the previous period. Again, the highest rate was observed for the over-70 age group. The first period was in line with our previous observations for the period between 1985 and 2007, highlighting a rising incidence of LOMG and a marked decrease in EOMG. During the COVID-19 period, incidence rates were lower in the first years whereas, when the pandemic ended, the previous trend was confirmed.

## 1. Introduction

Myasthenia gravis (MG) is an autoimmune disorder affecting the post-synaptic side of the neuromuscular junction, resulting in a characteristic muscle weakness [1,2]. MG patients are classified according to a combination of features: age of onset, i.e., early-onset (EOMG) and late-onset MG (LOMG); the affected muscle district, i.e., ocular (if symptoms remain confined to this region for at least 2 years), axial, limb, or bulbar; presence of specific auto-antibodies, i.e., mainly anti-acetyl-choline receptor (AchR, the most common), anti-muscle-specific kinase (MuSK), and anti-lipoprotein receptor-related peptide 4 (LRP4), abnormalities of the thymus (e.g., hyperplasia, thymoma), and other autoimmune disorders [3]. The frequency of MG varies in different estimates, and numerous population-based epidemiological studies on MG have been conducted globally since the 1950s [4,5,6,7]. A meta-analysis estimated a prevalence of 7.7 per 100,000 people and an incidence of 0.5 per 100,000 person-years [8]. In the last decade, the annual incidence rates of MG have ranged between 0.3 and 3.0 per 100,000, among the highest recently reported [9,10,11]. Since the mid-1980s, an incidence increase in the elderly [12,13,14] has been observed, i.e., in age groups where the diagnosis could be more challenging [15]. However, recent evidence has suggested that this incidence increase was attributable to an upward trend, particularly in LOMG [9,16,17,18]. Interestingly, a recent work has documented that the annual incidence of MG in Italy is among the highest in European countries [19,20], although, according to Antonini et al. [21], its incidence in this country has currently only been estimated by patients’ associations. Accordingly, epidemiological studies performed by our research group from 1985 to 2007 in the province of Ferrara, Northern Italy, indicated, in the last few years, an increase in the frequency of LOMG and a decrease in EOMG, suggesting a changing pattern of MG incidence [22,23]. Of note, between the end of 2019 and 2022, the world was hit by the COVID-19 pandemic, which had a profound impact on access to diagnosis and treatment [24,25]. Indeed, there were significant changes in the previously observed epidemiological measures that were linked to the reorganization of health services and reduced access to health facilities, and which were also related to the fear of contagion [26,27]. Coherently, recent work has shown a reduction in the incidence and hospitalization rate of MG in 2020, suggesting a role for COVID-19 in these changes [28]. Moreover, SARS-CoV-2 seems to trigger autoimmune processes due to severe inflammatory responses and interference with the immune system, causing a possible increase in the incidence of autoimmune diseases including both acute and chronic neurological disorders [29,30,31,32,33,34]. Consistently, several reports have revealed the onset of MG in patients who had been infected by SARS-CoV-2, hypothesizing a pathophysiological autoimmune link between these two entities [35,36,37,38]. However, evidence is still inconclusive, so further studies are needed to investigate the nature of this association.

This study aimed (i) to further clarify the trend of the incidence of MG in the area of Ferrara over an extended period of time, i.e., between 2008 and 2022; and (ii) to assess any epidemiological differences during the COVID-19 period (i.e., the period of 2019–2022) and investigate the relationship between SARS-CoV-2 infection and new diagnoses of MG in the same area.

## 2. Materials and Methods

### 2.1. Area of Investigation

The province of Ferrara is located in Emilia Romagna, a region in north-east Italy. The province is divided into three Local Health Districts (LHDs): Centre-North, West, and South-East (see Figure 1). For further characteristics about the study area, see our previous epidemiological studies [22,23].

The public healthcare system enables extensive access to data, encompassing both traditional paper records and computerized administrative databases. All patients with suspected MG are usually referred to the Outpatient Clinic for neuromuscular diseases of the Ferrara S. Anna University Hospital for a final diagnosis and follow-up. We assessed the MG incidence in the province of Ferrara between 1 January 2008 and 31 December 2018 and, separately, between 1 January 2019 and 31 December 2022. In the first period, a mean of 351,932 people per year lived in the province, 168,154 men and 182,878 women, while in the second period, a mean of 349,862 people per year lived in the province, 168,680 men and 181,182 women. From 2008 to 2022, the population underwent a gradual decrease because of the reduction in births throughout Italy, above all in the last three decades [data from the ISTAT dataset, which makes available the most recent official data on the population in Italian municipalities from surveys carried out at the Registry and Civil Status Offices of municipalities and from the Population Census, see https://demo.istat.it/ (accessed on 15 September 2023)] [39].

### 2.2. Case Selection

We adopted previous studies’ methodological approach [22,23] to have accurate comparisons. A complete enumeration approach was used by thoroughly investigating all the possible sources of MG cases: archives of the public and private neurologic and neurophysiologic units and services, of intensive care units, paediatrics, ophthalmologic thoracic surgery, and internal medicine departments (including both paper and computerized medical records). As in the past, we had the collaboration of general practitioners (GPs) that were employed in the study area. In order to verify the exhaustiveness of case collection, we examined all hospital discharges with a primary or secondary diagnosis of MG, codified 358.*, G70.*, or 8C60.* according to the International Classification of Diseases, 9th edition (ICD-9), 10th edition (ICD-10), or 11th edition (ICD-11), respectively. Finally, we screened the complete list of prescriptions for acetyl cholinesterase inhibitors. Following a comprehensive review of all medical records and the integration of data from diverse sources pertaining to the same individual, all diagnoses were cross-verified, resulting in, for the period of 2008–2018, an initial list of 131 cases. Of these, 14 with MG were living outside the province of Ferrara, whereas 11 had symptoms at onset that were compatible with MG but for which, at follow-up, an alternative diagnosis was confirmed (specifically, seven had a myopathy, three had a different neuromuscular disease, i.e., Lambert–Eaton myastheniform syndrome or congenital myasthenia, and one had a bulbar variant of motor neuron disease) and were therefore excluded from the analyses. The remaining 106 patients had onset of MG in the study area during the study period. Most of them (97) were diagnosed and/or followed by the neurologists of the Ferrara University Hospital. Nine subjects were also included because of their medical records and the information provided by their neurologists and GPs. Regarding the second period, on the other hand, the initial list of verified diagnoses included 32 patients, of whom 3 were excluded because they lived outside the study area, and thus, 29 patients had onset of MG in the study area during the study period and were taken into account in the analysis. All of them were diagnosed and/or followed by the neurologists of the Ferrara University Hospital. Table 1 summarizes the inclusion and exclusion criteria (see Table 1). The presence of serum anti-AchR antibodies was considered confirmatory, but their absence did not preclude inclusion if clinical and neurophysiological diagnostic criteria were fulfilled. For every included case, information on demographics, age of onset, dates of clinical onset and diagnosis, clinical characteristics, and classification based on criteria for standard clinical research of the Myasthenia Gravis Foundation of America (MGFA) [40] and results of diagnostic tests were all collected. Incidence was based on the year of diagnosis since, reasonably, this approach could represent the best trade-off for dealing with these pandemic emergency issues and in order to make comparisons between the two study periods. Indeed, in the first period, we confirmed, by means of accurate anamnestic evaluations and the information provided by GPs and/or territorial neurologists, that the clinical onset was in the same year of diagnosis (with an average time between onset of symptoms and diagnosis of 3 months). On the other hand, in the second period, the pandemic emergency made it more complex to acquire reliable anamnestic information (e.g., fewer visits to GPs and/or territorial neurologists) and, consequently, relying on the reported clinical onset could have introduced bias. Therefore, we chose to calculate the incidence based on the year of diagnosis in this second case as well. Furthermore, all patients included in the second period maintained a stable residence in the territory during the observation, thus fulfilling the inclusion criteria. Moreover, the subgroups were classified according to the following characteristics: the presence of thymoma; age of onset, i.e., EOMG and LOMG, < and ≥50 years, respectively; MGFA class, i.e., class I (ocular symptoms), II, III, IV (mild, moderate, and severe, respectively, generalized weakness), and V (intubation, with or without mechanical ventilation); the presence of specific auto-antibodies; the presence of other autoimmune diseases (e.g., thyroid disease, rheumatological disorders).

### 2.3. Statistical Analysis

An incidence case was defined as any patient with a diagnosis of MG residing in the study area in the period considered from 2008 until 2018 and from 2019 to 2022. The crude incidence estimates were sex- and age-standardized by direct method using the Italian (Italian National Institute of Statistics Population Census) and European [41] (http://ec.europa.eu/eurostat/data/database (2021), accessed on 15 September 2023) [42] population. Confidence intervals (CIs) for incidence rates at a 95% confidence level were computed assuming a Poisson’s distribution [43]. Due to the bimodal age pattern of MG incidence, we classified the patients into two categories, EOMG and LOMG, < and ≥50 years, respectively, as already stated [44,45,46]. To evaluate trends in incidence, we considered the results of our previous studies and estimated incidence rates for seven calendar periods: 1985–1990; 1991–1996; 1997–2002; 2003–2007; 2008–2012; 2013–2018; and 2019–2022. Continuous variables were reported as percentages or ratios, and comparisons were conducted using the chi-square test. A two-tailed probability distribution at the 95% confidence level was selected, with a significance level set at *p* < 0.05.

## 3. Results

### 3.1. First Period (2008–2018)

During the first period, a total of 106 cases (60 males and 46 females) were identified as new cases of MG, meeting the previously outlined criteria. All the patients were Italian Caucasian. The man/woman ratio was 1.3. The mean age at disease onset was 68.5 years (range 17–88 years, SD 14.3), 72.8 years for men (range 55–88 years, SD 8.4) and 62.8 years for women (range 17–87 years, SD 18.1). The mean crude annual incidence rate was 2.7 per 100,000 people (95%CI 2.3–3.3), 3.2 per 100,000 for men (95%CI 2.5–4.2) and 2.3 per 100,000 for women (95%CI 1.7–3.0), a non-significant sex difference (*p* > 0.05, overlapping CI). The incidence rates adjusted to the Italian population were 2.3 per 100,000 (95%CI 1.9–2.8), 2.6 per 100,000 (95%CI 2.0–3.4) for men and 2.1 per 100,000 (95%CI 1.5–2.7) for women, respectively. The incidence rates adjusted to the European population were 2.1 per 100,000 (95%CI 1.7–2.6), 2.3 per 100,000 (95%CI 1.8–3.0) for men and 1.9 per 100,000 (95%CI 1.4–2.6) for women. During the study period, rates increased: the mean annual incidence in the years 2008–2010 was 2.3 per 100,000 (95%CI 1.5–3.5), 2.4 per 100,000 (95%CI 1.7–3.3) in the years 2011–2014, and 3.4 per 100,000 (95%CI 2.5–4.5) in the years 2015–2018 (see Table 2). 

In this time frame, MG incidence showed a fluctuation: the lowest rate occurred in 2010, 1.9/100,000, and the highest in 2017–2018, with a rate of 3.7/100,000. The most elevated rates were noted in individuals aged over 70 years for both sexes. Over the entire observation period, the mean annual incidence was somewhat higher in rural areas (39 cases, mean population 121,896, average annual rate 2.9 per 100,000, 95%CI 2.1–4.0) than in urban areas (67 cases, mean population 229,137, average annual rate 2.6 per 100,000, 95%CI 2.0–3.4), with a non-significant difference (*p* > 0.05, overlapping CI). We detected 18 thymoma-associated MG cases (9 women and 9 men), 17% of the selected cases, giving an annual adjusted incidence rate of 0.5 per 100,000 (95%CI 0.3–0.4), 0.5 (95%CI 0.2–0.9) for men, and 0.4 (95%CI 0.2–0.9), a non-significant difference (*p* > 0.05, overlapping CI). Thymoma was found only in LOMG, both in men and women. Table 3 shows sex- and age-specific crude mean annual incidence rates (see Table 3).

The highest rates were observed in the age group of 70–79 years (8.4 per 100,000/year) for both sexes. We found 11 EOMG and 95 LOMG cases: all EOMG cases were women, while LOMG was more frequent in men, a significant sex difference (*p* < 0.05, non-overlapping CI). The incidence of the disease showed an upward trend for LOMG, while EOMG decreased in frequency over time. This trend concerns only non-thymoma MG cases. In accordance with the MGFA clinical categorization, the utmost severity during the pre-treatment stage was distributed as follows: 55.7% were classified as class I, 27.4%, 12.3%, 2.8%, and 1.9% as class II, III, IV, and V, respectively. No disparities were observed in the percentage of individuals within the classes between EOMG and LOMG (*p* > 0.05, overlapping CI). Anti-AchR antibodies were detected in 76.4% of patients, 74.7% of men and 82.6% of women, 87.1% in EOMG, and 86.7% in LOMG, without any significant difference over the observation period (*p* > 0.05, overlapping CI). No other auto-antibodies (e.g., anti-MuSK, anti-LRP4) were found in the patients included. A total of 33% of women with EOMG had other autoimmune onset diseases: thyroiditis, psoriatic arthritis, undifferentiated connective tissue disease, and systemic lupus erythematosus (SLE). Among the cases with LOMG, we found autoimmune diseases, above all thyroiditis, in 18% of women’s cases.

### 3.2. Second Period (2019–2022, i.e., COVID-19 Period)

In the second period, 29 cases (16 men and 13 women) were detected as new cases of MG fulfilling the criteria described above. All the patients were Italian Caucasian, except one, who was a child from Eastern Europe. The man/woman ratio was 1.2. The mean age at disease onset was 68.0 years (range 2–88 years, SD 17), 72.2 years for men (range 53–88 years, SD 11.8) and 62.7 years for women (range 2–84 years, SD 21.1). The mean crude annual incidence rate was 2.1 per 100,000 (95%CI 1.4–3.0), 2.4 per 100,000 for men (95%CI 1.4–3.9) and 1.8 per 100,000 for women (95%CI 1.0–3.1), with a non-significant sex difference (*p* > 0.05, overlapping CI). The incidence rates adjusted to the Italian population were 1.8 per 100.000 (95%CI 1.2–2.5), 2.0 per 100,000 (95% CI 1.2–3.3) for men and 1.5 per 100,000 (95%CI 0.8–2.6) for women, respectively. The incidence rates adjusted to the European population were 0.9 per 100,000 (95%CI 0.6–1.3), 0.9 per 100,000 (95%CI 0.5–1.5) for men and 0.8 per 100,000 (95%CI 0.4–1.4) for women. When compared with the first period, the mean annual incidence showed a marked decrease, revealing not only a downward trend (compared to the upward trend shown in the period of 2008–2018), but also values that were markedly (though not statistically significant, *p* > 0.05, overlapping CI) lower than the lowest value found previously, i.e., 2.3 in the period of 2008–2010 (see Table 4).

Also in this timeframe, the MG incidence showed a fluctuation: the lowest rate occurred in 2019, 0.9/100,000, and the highest in 2022, with a rate of 4.7/100,000. The most elevated rates were noted in individuals aged over 70 years for both sexes. Over this observation period, the mean annual incidence was somewhat greater in urbanized areas (20 cases, mean population 225,836.30, average annual rate 2.2 per 100,000, 95%CI 1.3–3.4) than in rural regions (9 cases, mean population 117,084.20, average annual rate 1.9, 95%CI 0.9–3.6), with a non-significant difference (*p* > 0.05, overlapping CI). We found two thymoma-associated MG cases (one man and one woman), 6.9% of the selected cases, giving an annual adjusted incidence rate of 0.1 per 100,000 (95%CI 0.02–0.5), 0.1 (95%CI 0.004–0.8) for men, and 0.14 (95%CI 0.003–0.8), a non-significant difference (*p* > 0.05, overlapping CI). Thymoma was found only in late-onset MG, both in men and women. Table 5 summarizes the sex- and age-specific crude mean annual incidence rates (see Table 5).

The highest rates were observed in the age group of 80–89 years (7.4 per 100,000/year), but only for men, as the highest rate in women was in the 70–79 age group. We found just 1 EOMG and 28 LOMG cases: the early onset of the disease before 50 years of age was found exclusively in a female child (two years of age), and LOMG was more frequent in men, a significant sex difference (*p* < 0.05, non-overlapping CI). Also in this time period, the incidence of the disease showed an upward trend for LOMG, while EOMG decreased in frequency over time. Such a trend concerns only non-thymoma MG cases (the inclusion of only two patients with MG associated with thymoma prevents similar trends from being assessed in this group as well). In accordance with the MGFA clinical categorization, the utmost severity during the pre-treatment stage was subdivided as follows: 44.8% were classified as class I, 17.2%, 37.9%, 0%, and 0% as class II, III, IV, and V, respectively. The only EOMG in this group was in class III of the MGFA clinical classification. Table 6 shows the comparison of the distribution of patients according to MGFA class between the two periods considered (see Table 6):

Anti-AchR antibodies were detected in 82.8% of patients, 87.5% of men and 76.9% of women, all in LOMG, as the only case of EOMG was negative for all antibodies investigated. Also in this period, no other auto-antibodies (e.g., anti-MuSK, anti-LRP4) were found in the patients included. In total, 61.5% of women with LOMG had other autoimmune diseases: thyroiditis (38.5%), undifferentiated connective tissue disease (7.7%), SLE (7.7%), and bronchial asthma (7.7%), and two women, in addition to thyroiditis, also suffered from autoimmune hepatopathy and coeliac disease, respectively (7.7% of the total sample each). Only one man with LOMG was found to have autoimmune thyroiditis (6.2%). The only EOMG patient had no autoimmune co-pathologies. Of note, seven of the included patients were infected by SARS-CoV-2 (24.1%), six LOMG (85.7%) and one EOMG (14.3%), five women (71.4%) and two men (28.6%), respectively. Interestingly, three of the women contracted COVID-19 before the onset and diagnosis of the disease: one was the only case of EOMG and contracted the infection 2 months before the onset of MG symptoms, while the remaining ones contracted it 4 and 12 months before the onset of MG, respectively. The other four cases of COVID-19 occurred after the onset and diagnosis of MG. Regarding the COVID-19 vaccines, only two patients (7.0%, both women, one of whom was the EOMG case) did not receive any vaccine doses, while the other subjects received at least three doses. Among these, eight men (27.6%) and six women (20.7%) received the first and the second dose before the onset of MG. The latency interval between the two events (i.e., vaccination and MG onset) was more than 6 months for all but two women, who received the COVID-19 vaccine one month and two months before MG onset, respectively. In contrast, the other subjects (eight men, 27.6%, and five women, 17.2%) received the first dose of the vaccine after the onset of myasthenic symptoms.

## 4. Discussion

We will discuss the results obtained from both periods considered.

### 4.1. Time Period of 2008–2018

The mean annual MG incidence in the study area for the years 2008–2018 was 2.7 per 100,000 population, among the highest ever detected [9,10,11,16,47]. Of note, this incidence rate is extremely similar (i.e., 2.8/100,000) to that documented in a recent retrospective study conducted in Germany by the group of Wartmann et al. [28], showing a sharp rise in incidence of MG in Europe, especially in the elderly (i.e., LOMG) [9,16]. The descriptive incidence analysis was conducted using a complete enumeration approach on a clearly defined population of approximately 350,000 people, ensuring precise and thorough case compilation while minimizing the potential for bias. Considering the structure of healthcare services in Ferrara, coupled with the meticulous methodological approach to case identification and the recurrent surveys conducted in the same area by the same investigators, any under-ascertainment was expected to be minimal. Every individual enrolled in this study underwent an extensive neurological follow-up, ensuring accurate diagnosis, especially for mild and seronegative cases. This marks the third examination of MG incidence in the province of Ferrara, with the same team of researchers conducting the initial study between 1985 and 2000 [23], followed by another investigation spanning 2001 to 2007 [22]. The mean incidence rates during those periods remained substantially stable (i.e., between 1.8 and 2.0 per 100,000), without any significant temporal trend. Since 2000, a significant increase in the incidence of LOMG, above all among men, and a decrease in EOMG has been detected (see Figure 2); in particular, the findings were related to non-thymoma MG.

A recent improvement in the diagnosis of MG does not solely justify the reported incidence increase, above all in LOMG rates, also if we consider the decrease in EOMG frequency: an underestimation of young MG cases in recent years seems unlikely. In this survey, our results still confirm that differences in estimates at different times can reflect, to some extent, a changing epidemiological trend, with an increase in LOMG and a decrease in EOMG. While the rise in late-onset incidence is a confirmatory result also in Asia and Africa [12,13,18,22,48,49,50], as far as we know, the decrease in early-onset non-thymoma MG has not been reported previously, particularly in European populations, which are still known to have a significantly lower incidence of EOMG (especially in youth) in comparison with Asian countries [51,52]. As described elsewhere in the last decade worldwide [8,9,11,16,17,18,53], in our investigation, the incidence of MG in the years 2008–2018 reaches its peak between the age of 60 and 80, particularly in men compared to women. Our results only refer to autoimmune, non-thymoma-associated MG, considering that the frequency of thymoma-related, paraneoplastic MG has not substantially changed across different time periods or at any age. The decrease in incidence rates in EOMG affecting women and the rising incidence of late-onset MG among men are confirmatory results of our previous epidemiological observations [22,23], which lead us to consider MG a disease of the elderly, indicating the necessity for further investigation into the impact of external environmental factors in MG etiology [5,54]. Moreover, we can support the hypothesis that thymoma-associated MG and early- and late-onset non-thymoma MG should be regarded as distinct categories [12,13,55], determined by different environmental risk factors and by different immunogenetic backgrounds [5,22,54].

### 4.2. Time Period 2019–2022, i.e., COVID-19 Period

The mean annual MG incidence in the study area for the years 2019–2022 was 2.1 per 100,000, a substantial decrease from the first period. Indeed, although the population in the area showed a downward trend throughout the study period, the decline is not likely to justify such an abrupt reduction compared to the previous decade. Moreover, the incidence rates were particularly low in 2019, 2020, and 2021 (respectively, 0.9/100,000, 1.4/100,000, and 1.5/100,000), while a marked increase was observed in 2022, equal to 4.7/100,000, which is higher than the highest value recorded in previous decades of observation (i.e., 3.4/100,000 in the period of 2015–2018). These data also show a clear fluctuation between the beginning and end of the second study period. Since the population did not change significantly over this time span, it is reasonable to assume that the differences observed can be attributed to the impact of the COVID-19 pandemic. Specifically with regard to MG, in Germany, Wartmann et al. [28] also documented a significant decrease in its incidence at the beginning of the COVID-19 period compared to the past, which was hypothesized to be attributable to the spread of SARS-CoV-2. This interpretation is further supported by the fact that, in 2022, there was a previously unprecedented peak in incidence, which can probably be attributed not only to the incidence of new MG cases, but also to the diagnoses that had not been made in the previous two years. For example, the group of Van den Bulck [56] observed that, following the pandemic, the incidence of chronic diseases in Flanders decreased and returned to baseline values at the end of the emergency, and this could also be the case for the changes observed by our group. Despite these considerable limitations, interestingly, the prevalence of non-thymoma MG and LOMG is also confirmed in this period; they are significantly more frequent than thymoma MG cases (*n* = 2, 6.9% of new cases in the period) and EOMG (*n* = 1, 3.4% of new cases in the period). It is unlikely that these data are underestimated in terms of the included patients, as all patients completed the regular diagnostic work-up, which confirmed the presence of thymoma in only two cases. In addition, limitations in access to care, for the reasons already discussed, particularly affected the elderly population, whereas young subjects, who are less exposed to the more serious consequences of SARS-CoV-2 infection, had a less significant limitation of hospital access [57,58,59]. Therefore, in this sense, the trend already observed in the previous period is confirmed.

However, another intriguing possibility is that the increase in incidence recorded in 2022 is attributable, at least in a small part of the sample, to the SARS-CoV-2 infection or related vaccination. Indeed, substantial evidence in the literature has suggested a correlation between COVID-19 and the triggering of autoimmune diseases [29,30], including those affecting the nervous system [33,34]. Consistently, after COVID-19, there was an increase in several diseases that are linked to an autoimmune etiopathogenesis [31,60], and this could also be the case for some of our patients, in particular a two-year-old female, who apparently had no other known risk factors and developed MG two months after SARS-CoV-2 infection, as well as a woman with an onset of LOMG a few months after COVID-19 disease. This argument is also applicable to MG, since it is an autoimmune disease that could be influenced by the massive release of inflammatory cytokines related to COVID-19, leading to immune-mediated damage, as has already been suggested by numerous cases reported in the literature [36,37,38]. Some recent literature reviews also suggest a link between autoimmune disorders and SARS-CoV-2 vaccination [61,62]. In our case, all but two of the included patients were vaccinated against SARS-CoV-2; however, although 14 of them received two doses of vaccine before MG onset, the long latency (i.e., more than 6 months) between the two events makes a direct causal link unlikely. These considerations are applicable for all patients except for two women, who received the second dose one month and two months before the onset of myasthenic symptoms, respectively. Interestingly, the cases reported in the literature of newly diagnosed MG after COVID-19 vaccination also occurred after the second dose, albeit at variable time intervals, generally a few days or weeks [63,64,65,66]. However, it is possible, as has recently been speculated, that vaccination only contributed to the manifestation of sub-clinical myasthenic symptoms and consequently led to the recognition and diagnosis of MG [67]. Further studies, also with longer follow-ups, will be needed to assess the real impact of COVID-19 and its vaccination on autoimmune diseases.

Although this second period offered us a unique opportunity to examine the impact of SARS-CoV-2 pandemics on the incidence of MG in the province of Ferrara by comparing it with previous decades, already evaluated by our research group, it should be noted that the aforementioned limitations related to the pandemic emergency likely made the incidence data collected in the second period less reliable. Moreover, hypotheses about a link between SARS-CoV-2 infection or vaccination and MG onset, given the extremely small sample of patients, remain highly speculative suggestions based on evidence that is already present in the literature, but much larger study samples would be needed to ascertain these correlations. Therefore, our group will continue the follow-up in the coming years to assess whether the trends return to align with those prior to the COVID-19 period or, if they do not, the underlying reasons will be analysed.

## 5. Conclusions

We conducted an extensive observational study on the incidence of MG in the province of Ferrara (Northern Italy). The results for the period of 2008–2018 were in line with previous observations of our group, highlighting in particular a decrease in the incidence rates in EOMG and a rising incidence of LOMG, leading us to see MG as a disease of the elderly. Moreover, since the incidence of thymoma-related MG has not changed over time at any age, we suggest that thymoma-associated MG and non-thymoma EOMG and LOMG should be regarded as distinct categories, determined by different environmental risk factors and by different immunogenetic backgrounds. Concerning the COVID-19 period, as was reasonable to expect due to the pandemic emergency and its consequences, incidence rates were reduced compared to the previous period, in particular those of EOMG and thymoma-related incidence rates, especially in the early years of the COVID-19 pandemic. Of note, in 2022, when the emergency came to an end, there was a sharp increase in incidence, which not only confirmed the upward trend of previous decades but may also have been related to undiagnosed cases from the period of SARS-CoV-2-associated limitations. Our group will continue the observation in the coming years, so as to assess the trends in the incidence of MG at a later point from the COVID-19 pandemic.

## Figures and Tables

**Figure 1 jcm-13-00236-f001:**
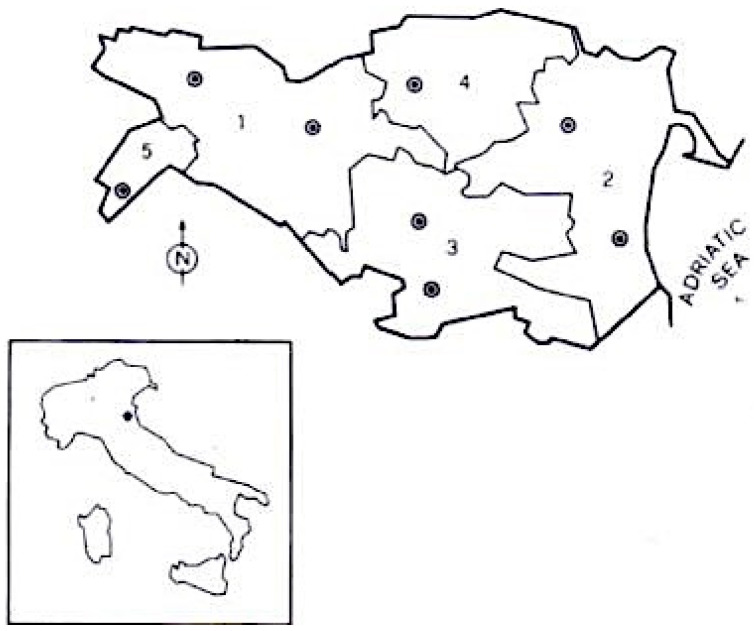
Province of Ferrara (Italy), area of investigation. Borders of the various public health districts are indicated (1 = Ferrara; 2 = Comacchio; 3 = Portomaggiore; 4 = Copparo; 5 = Cento; • = neurological centres).

**Figure 2 jcm-13-00236-f002:**
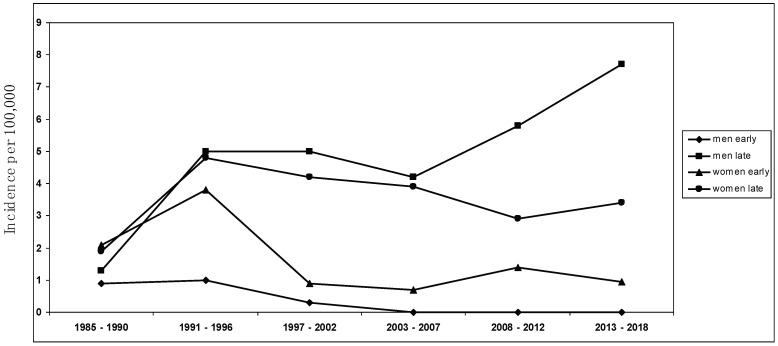
Sex-specific incidence trend over the study period from 1985 to 2018 ([22,23], present study).

**Table 1 jcm-13-00236-t001:** Summary of the inclusion and exclusion criteria.

Inclusion Criteria	Exclusion Criteria
MG diagnosis according to clinical (typical history of fluctuating muscle weakness, clinical evidence of abnormal fatigability relieved by rest) and neurophysiological features (abnormal single-fibre electromyogram or typical response at low-frequency repetitive stimulation), confirmed by an experienced neurologist of the Outpatient Clinic for neuromuscular diseases and by a positive response to MG-related treatment [3]	Confirmation of alternative diagnosis
Living in the study area at the time of clinical onset	Living outside the study area at the time of clinical onset

**Table 2 jcm-13-00236-t002:** MG incidence in the province of Ferrara and mean rate in the years 2008–2018.

	Population			Cases			Mean Rates per 100,000 (CI 95%)		
	Total	Men	Women	Total	M	F	Total	Men	Women
2008	355,089	169,703	185,386	10	3	7	2.8	1.8	3.8
2009	357,980	171,399	186,581	8	3	5	2.2	1.7	2.7
2010	358,972	171,695	187,277	7	6	1	1.9	3.5	0.5
2008–2010	357,347	170,932	186,415	25	11	14	2.3 (1.5–3.5)	2.3 (1.2–4.2)	2.3 (1.3–3.9)
2011	359,994	171,912	188,082	10	7	3	2.8	4.1	1.6
2012	352,856	168,047	184,809	7	1	6	2.0	0.6	3.2
2013	352,723	168,281	184,442	9	6	3	2.5	3.6	1.6
2014	355,101	169,587	185,514	8	5	3	2.2	2.9	1.6
2011–2014	355,169	169,457	185,712	34	19	15	2.4 (1.7–3.3)	2.8 (1.7–4.4)	2.0 (1.1–3.3)
2015	354,073	169,208	184,865	10	5	5	2.8	2.9	2.7
2016	351,437	168,148	183,289	11	8	3	3.1	4.8	1.6
2017	348,362	166,883	181,479	13	8	5	3.7	4.8	2.8
2018	346,975	166,605	180,370	13	8	5	3.7	4.8	2.8
2015–2018	350,212	167,711	182,501	47	29	18	3.4 (2.5–4.5)	4.3 (2.9–6.2)	2.5 (1.5–3.9)
2008–2018	351,032	168,154	182,878	106	60	46	2.7 (2.3–3.3)	3.2 (2.5–4.2)	2.3 (1.7–3.0)

**Table 3 jcm-13-00236-t003:** Sex-and age-specific incidence rates (per 100,000) of MG in the province of Ferrara, 2008–2018.

	Mean Population			Cases			Incidence Rates per 100,000		
Age Group	Total	Men	Women	Total	Men	Women	Total	Men	Women
0–19	50,172	25,852	24,319	1	0	1	0.2	/	0.4
20–29	29,822	15,261	14,561	3	0	3	0.9	/	1.9
30–39	45,085	22,817	22,268	1	0	1	0.2	/	0.4
40–49	55,475	27,867	27,608	6	0	6	1.0	/	2.0
50–59	52,982	25,811	27,171	11	5	6	1.9	1.8	2.0
60–69	47,925	22,629	25,296	24	16	8	4.5	6.4	2.9
70–79	41,054	18,199	22,855	38	25	13	8.4	12.5	5.2
Over 80	28,878	10,076	18,802	22	14	8	6.9	12.6	3.9

**Table 4 jcm-13-00236-t004:** MG incidence in the province of Ferrara and mean rate in the years 2019–2022 (COVID-19 era).

	Population			Cases			Mean Rates per 100,000 (CI 95%)		
	Total	Men	Women	Total	M	F	Total	Men	Women
2019	345,538	166,174	179,364	3	1	2	0.9	0.6	1.1
2020	344,510	165,912	178,598	5	4	1	1.4	2.4	0.6
2021	342,061	165,003	177,058	5	4	1	1.5	2.4	0.6
2022	339,573	164,303	175,270	16	7	9	4.7	4.3	5.1
2019–2022	342,920.5	165,348	177,572.5	29	16	13	2.1 (1.4–3.0)	2.4 (1.4–3.9)	1.8 (1.0–3.1)

**Table 5 jcm-13-00236-t005:** Sex-and age-specific incidence rates (per 100,000) of MG in the province of Ferrara, 2019–2022.

	Mean Population			Cases			Incidence Rates per 100,000		
Age Group	Total	Men	Women	Total	Men	Women	Total	Men	Women
0–49	541,045.5	275,339.75	265,705.75	1	0	1	0.05 (0.001–0.3)	0	0.09 (0.002–0.05)
50–59	56,980	28,063.25	28,916.75	7	3	4	3.1 (1.2–6.3)	2.7 (0.5–7.8)	3.5 (0.9–8.9)
60–69	48,606.75	23,003	25,603.75	5	3	2	2.6 (0.8–6.0)	3.3 (0.7–9.7)	1.9 (0.2–7.0)
70–79	41,255.5	18,843.25	22,412.25	8	4	4	4.8 (2.1–9.5)	5.3 (1.4–13.6)	4.5 (1.2–11.4)
80–89	26,843	10,565.25	16,277.75	8	6	2	7.4 (3.2–14.7)	14.2 (5.2–31.0)	3.1 (0.4–11.1)
90+	5857.25	1611	4246.25	0	0	0	0	0	0

**Table 6 jcm-13-00236-t006:** Distribution of patients according to MGFA classes in the two periods.

MGFA Class	2008–2018	2019–2022
I	59	13
II	29	5
III	13	11
IV	3	0
V	2	0

## Data Availability

The dataset used for the data analysis is available on reasonable request to the Corresponding Author.

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
