# Peer review of "The Incidence of Myasthenia Gravis in the Province of Ferrara, Italy, in the Period of 2008–2022: An Update on a 40-Year Observation and the Influence of the COVID-19 Pandemic"

_jcm, 2023, doi:10.3390/jcm13010236_

Round 1

Reviewer 1 Report

Comments and Suggestions for Authors

think the paper is well written. It brings important data about MG epidemiology before and after pandemics in this region.

I would just suggest to make more clear in the methods the classification in subgroups.

Author Response

Think the paper is well written. It brings important data about MG epidemiology before and after pandemics in this region.

We thank the Reviewer for the positive feedback on our submission.

I would just suggest to make more clear in the methods the classification in subgroups.

We thank the Reviewer for emphasizing this point, we agree. We have added clarification about the classification in subgroups to the main manuscript (please see Methods section, paragraph 2.2. Case selection):

Moreover, the subgroups were classified according to the following characteristics: the presence of thymoma; age of onset, i.e. EOMG and LOMG, < and ≥ 50 years respectively; MGFA class, i.e. class I (ocular symptoms), II, III, IV (mild, moderate, and severe, respectively, generalized weakness), and V (intubation, with or without mechanical ventilation); the presence of specific auto-antibodies; the presence of other auto-immune diseases (e.g. thyroid disease, rheumatological disorders).”

Reviewer 2 Report

Comments and Suggestions for Authors

This manuscript is a complete survey of MG patients in the province of Ferrara, Northern Italy, over two timeframes (2008-18 and 2019-22, i.e., Covid-19 pandemics) and considered EOMG, LOMG, thymoma, and non-thymoma MG.

                 The study population is small. Therefore, the content of the manuscript is descriptive.

1.     Introduction: Line 44. “carcinoma” should be replaced by thymoma.

2.     Materials and Methods: Please provide the total population of Ferrara, along with the era.

3.     Materials and Methods: Please provide patients' inclusion and exclusion criteria in a table format.

4.     Results: Please provide the statistical values of the analyses. It isn't easy to understand whether the observation is meaningful.

5.     Figure 2: The markers of the graph are too small to distinguish. Please use color markers or larger ones.

6.     The effects of the COVID-19 pandemic or vaccination against COVID-19 are critical points in this manuscript. However, the authors did not show solid data to discuss those effects on MG incidence. Please provide the observations to support the authors’ hypothesis.

Author Response

This manuscript is a complete survey of MG patients in the province of Ferrara, Northern Italy, over two timeframes (2008-18 and 2019-22, i.e., Covid-19 pandemics) and considered EOMG, LOMG, thymoma, and non-thymoma MG. The study population is small. Therefore, the content of the manuscript is descriptive.

1) Introduction: Line 44. “carcinoma” should be replaced by thymoma.

We thank the Reviewer for the constructive feedback on our submission. We agree, thus we modified the introduction according to your suggestion (please see the Introduction section). Specifically:

“abnormalities of the thymus (e.g. hyperplasia, thymoma),”

2) Materials and Methods: Please provide the total population of Ferrara, along with the era.

We thank the Reviewer for emphasizing this point. Please note that we have already reported these data, as well as in Table 2 and 4, in the Methods section, section 2.1. Area of investigation. Specifically, please see:

“We assessed the MG incidence in the province of Ferrara between January 1, 2008 and December 31, 2018 and, separately, between January 1, 2019 and December 31, 2022. In the first period, a mean of 351,932 people per year lived in the province, 168,154 men and 182,878 women, while in the second period, a mean of 349,862 people per year lived in the province, 168,680 men and 181,182 women.”.

3) Materials and Methods: Please provide patients' inclusion and exclusion criteria in a table format.

We thank the Reviewer for this suggestion, which allowed us to make our paper clearer and smoother. Based on your suggestion, we have added Table 1 and its reference in the text to the Methods section, paragraph 2.2 Case selection (please see the Main manuscript).

“Table 1 summarises the inclusion and exclusion criteria (see Table 1).”

4) Results: Please provide the statistical values of the analyses. It isn't easy to understand whether the observation is meaningful.

We thank the Reviewer for emphasizing this point, we agree. To make this clearer, we have reported in brackets whether the p-value is greater or less than 0.05. Please note that we have evaluated confidence intervals (CI) to check this aspect. Specifically, if the CI were disjointed, the differences between the two compared rates were defined as 'statistically significant' (not due to chance). Conversely, if the CI had an overlap, even partial, this indicated that the differences were considered 'not statistically significant'. Please see the main manuscript, in the Result section.

5) Figure 2: The markers of the graph are too small to distinguish. Please use color markers or larger ones.

We thank the Reviewer for this suggestion, which allowed us to improve the quality of our figure. We modified Figure 2 according to your indications, specifically we enlarged the figure in order to make the marker clearer (please see Figure 2 in the Main manuscript).

6) The effects of the COVID-19 pandemic or vaccination against COVID-19 are critical points in this manuscript. However, the authors did not show solid data to discuss those effects on MG incidence. Please provide the observations to support the authors’ hypothesis.                 

We thank the Reviewer for this valuable suggestion, which allowed us to make our work clearer. Please note that our paper is a descriptive epidemiological study in which we studied the incidence of MG over the study period. In addition, we also evaluated the period of the Covid-19 pandemic and observed any differences from previous periods. Specifically, we divided the second section of our Discussion (i.e. paragraph 4.2. Time-period 2019-2022) into two parts: in the first one, we described the incidence rates of the Covid-19 period and analysed the differences from the first study period, also considering relevant works in the literature on the subject. Instead, in the second part, we evaluated the aspect of Sars-Cov2-infection and vaccination in patients newly diagnosed with MG. Please note that we only suggested reasonable explanations but just as hypotheses, since the patient sample is small and our study is descriptive, thus we hypothesised the possible presence of a link between Sars-Cov2 and MG onset based on evidence already available in the literature. However, we agree with the limitations described by the Reviewer and, thus, we have included a sentence in the Limitations of the second study period (please see Discussion section, paragraph 4.2.):

“Moreover, hypotheses about a link between Sars-Cov2 infection or vaccination and MG onset, given the extremely small sample of patients, remain highly speculative suggestions based on evidence already present in the literature, but much larger study samples would be needed to ascertain these correlations.”

Reviewer 3 Report

Comments and Suggestions for Authors

This is an interesting study examining the incidence of myasthenia gravis (MG) in an Italian province in the period 2008-2022. The research group is leading in European neuroepidemiology, and the province has established routines for epidemiological collection of cases.

1: The manuscript should be shortened. This is especially true for the Introduction and for "Area of investigation", but also other parts should be condensed. They should concentrate on the MG epidemiology,

2: The title is misleading.  In this article, they include patients in the time period 2008-2022.

3: They generally give figures with two decimals. With the number of patients being 29 and 106 in the main groups, this is misleading and implies a false accuracy. In subgroups there are even fewer patients. 

4: In the Methods, they should state more clearly that onset is defined as symptom onset, not time of diagnosis. 

5: Only 11 patients were excluded because of symptom appearance before 2008. Why so few?

6: Conclusions should be strict and reliable. What is "highly speculative hypotheses" have no place in a conclusion and should be removed.  

Author Response

This is an interesting study examining the incidence of myasthenia gravis (MG) in an Italian province in the period 2008-2022. The research group is leading in European neuroepidemiology, and the province has established routines for epidemiological collection of cases.

We thank the Reviewer for the positive feedback on our submission. 

1) The manuscript should be shortened. This is especially true for the Introduction and for "Area of investigation", but also other parts should be condensed. They should concentrate on the MG epidemiology.

We thank the Reviewer for emphasizing this point, we agree. Therefore, we revised our manuscript and shortened and summarised all the parts that were less relevant or referred to the previous work of our group for details already provided. Specifically, we focused on the Introduction and the paragraph on the study area, but the whole work has been thoroughly revised in order to address your advice. Please see the main manuscript.

2) The title is misleading. In this article, they include patients in the time period 2008-2022.

We thank the Reviewer for the valuable suggestion, we agree. We wished to emphasise the fact that we have updated the data from our group's previous observations, but to avoid misinterpretation we have changed the title as follows:

Incidence of myasthenia gravis in the province of Ferrara, Italy, in the period 2008-2022: an update on a 40-year observation and the influence of Covid-19 pandemics”

3) They generally give figures with two decimals. With the number of patients being 29 and 106 in the main groups, this is misleading and implies a false accuracy. In subgroups there are even fewer patients. 

We thank the Reviewer for the important suggestion, we agree. Thus, we revised the main manuscript and the tables in order to report only decimal (please see the Abstract, the Main manuscript and Tables).

4) In the Methods, they should state more clearly that onset is defined as symptom onset, not time of diagnosis. 

We thank the Reviewer for emphasizing this point. Please note that we chose to estimate the incidence according to the time of diagnosis because, in the second study period, it is much more complex to be sure in dating the clinical onset of symptoms. However, please note that, in the first study period, we calculated the incidence according to the year of diagnosis but we also confirmed (through careful anamnestic investigation and the collaboration of general practitioners and/or neurologists in the area) that the symptom onset fell in the same year of diagnosis with an average time between onset and diagnosis of about 3 months. On the contrary, in the second period (i.e. the Covid period), although we tried to adopt the same methodological approach in terms of rigour, the detailed information we collected in the first period was not as readily available (e.g. less accessibility to general practitioners and territorial neurologists), so we thought that adopting an approach based on the year of diagnosis might be the best trade-off for dealing with these pandemic emergency issues and in order to make comparisons between the two periods. In the manuscript, in the Methods section, paragraph 2.2 Case selection, we added this important clarification as follows:

“Incidence was based on the year of diagnosis since, reasonably, this approach could represent the best trade-off for dealing with these pandemic emergency issues and in order to make comparisons between the two study periods. Indeed, in the first period, we confirmed, by means of accurate anamnestic evaluations and the information provided by GPs and/or territorial neurologists, that the clinical onset was in the same year of diagnosis (with an average time between onset of symptoms and diagnosis of 3 months). On the other hand, in the second period, the pandemic emergency made it more complex to acquire reliable anamnestic information (e.g. fewer visits to GPs and/or territorial neurologists) and, consequently, relying on the reported clinical onset could have introduced bias. Therefore, we chose to calculate the incidence on the year of diagnosis in this second case as well. Furthermore, all patients included in the second period maintained a stable residence in the territory during the observation, thus fulfilling the inclusion criteria.”.

5) Only 11 patients were excluded because of symptom appearance before 2008. Why so few?

We really thank the Reviewer for pointing out this error. We made a mistake in the wording, we actually meant that some of the patients initially had symptoms compatible with MG, but for whom an alternative diagnosis was confirmed at follow-up. Specifically, in the manuscript we revised as follows (please see Methods section, paragraph 2.2 Case selection):

“whereas 11 had symptoms at onset that were compatible with MG but for which, at follow-up, an alternative diagnosis was confirmed (specifically, seven had a myopathy, three had a different neuromuscular disease, i.e. Lambert-Eaton myastheniform syndrome or congenital myasthenia, and one had a bulbar variant of motor neuron disease) and were therefore excluded from the analyses.”

6) Conclusions should be strict and reliable. What is "highly speculative hypotheses" have no place in a conclusion and should be removed. 

We thank the Reviewer for emphasizing this point, we agree. We revised our conclusions based on your suggestions.

Reviewer 4 Report

Comments and Suggestions for Authors

This study evaluated MG incidence rate in the province of Ferrara, Northern Italy, and assessed its possible relationship with Sars-CoV2 infection or Covid-19 vaccination. However, this study did not use the medical record registration system to ensure the authenticity and integrity of the data, and the total number of patients included was small. The data in this study are not representative enough.

Comments on the Quality of English Language

minor revision for language

Author Response

This study evaluated MG incidence rate in the province of Ferrara, Northern Italy, and assessed its possible relationship with Sars-CoV2 infection or Covid-19 vaccination. However, this study did not use the medical record registration system to ensure the authenticity and integrity of the data, and the total number of patients included was small. The data in this study are not representative enough.

We thank the Reviewer for the constructive feedback on our submission. Please note that this is a descriptive epidemiological study in which we described the incidence of MG during the study period and its characteristics (and any differences) also during the Covid-19 pandemics. In addition, we adopted a complete enumeration approach to make sure that we included all patients who met the inclusion and exclusion criteria which, reasonably, accurately describe the incidence of MG in the study area during the periods of interest. Our study group has been conducting epidemiological studies in this area for about 50 years and we hope that our experience, together with the methodological rigour adopted, allowed us to identify all patients to assess the MG incidence. Specifically, this methodological approach has already been used by our group (see Casetta et al., 2004, 2010) and the decision to adopt such a rigorous and already validated strategy allowed us to obtain reliable data and to ensure comparability with our group's previous observations periods. However, we clarified as follows (please see Methods section, 2.2. Case selection):

“A complete enumeration approach was used by thoroughly investigating all the possible sources of MG cases: archives of the public and private neurologic and neurophysiologic units and services, of intensive care units, paediatrics, ophthalmologic thoracic surgery, and internal medicine departments (including both paper and computerized medical records). As in the past, we had the collaboration of general practitioners (GPs) employed in the study area. In order to verify the exhaustiveness of case collection, we examined all hospital discharges with a primary or secondary diagnosis of MG, codified 358.*, G70.* or 8C60.* according to the International Classification of Diseases, 9th edition (ICD-9), 10th edition (ICD-10) or 11th edition (ICD-11), respectively. Finally, we screened the complete list of prescriptions for acetyl cholinesterase inhibitors. After examining all the medical documents and the linking of data from various sources regarding the same subject, all diagnoses were verified”.

Concerning the Covid-19 period, as already mentioned, we described the MG rates also in this timeframe and observed any differences compared to previous periods, suggesting reasonable explanations but only in the form of hypotheses, as we agree with the Reviewer about the limitations of the reduced sample. However, our aim was only to update the observation of previous decades in our study area (including the changes in the pandemic period) and to present data as accurate as in the past. However, since we agree with the Reviewer on the above-mentioned limitations, we have included a sentence in the Limitations of the second study period (please see Discussion section, paragraph 4.2.):

“Moreover, hypotheses about a link between Sars-Cov2 infection or vaccination and MG onset, given the extremely small sample of patients, remain highly speculative suggestions based on evidence already present in the literature, but much larger study samples would be needed to ascertain these correlations.”